# An Insight into the Essential Role of Carbohydrate-Binding Modules in Enzymolysis of Xanthan

**DOI:** 10.3390/ijms24065480

**Published:** 2023-03-13

**Authors:** Xin Ni, Tong Fu, Xueyan Wang, Jingjing Zhao, Zhimin Yu, Xianzhen Li, Fan Yang

**Affiliations:** School of Biological Engineering, Dalian Polytechnic University, Dalian 116034, China

**Keywords:** xanthan, endotype xanthanase, carbohydrate-binding modules, enzymolysis

## Abstract

To date, due to the low accessibility of enzymes to xanthan substrates, the enzymolysis of xanthan remains deficient, which hinders the industrial production of functional oligoxanthan. To enhance the enzymatic affinity against xanthan, the essential role of two carbohydrate binding modules—*Mi*CBMx and *Psp*CBM84, respectively, derived from *Microbacterium* sp. XT11 and *Paenibacillus* sp. 62047—in catalytic properties of endotype xanthanase *Mi*Xen were investigated for the first time. Basic characterizations and kinetic parameters of different recombinants revealed that, compared with *Mi*CBMx, *Psp*CBM84 dramatically increased the thermostability of endotype xanthanase, and endowed the enzyme with higher substrate affinity and catalytic efficiency. Notably, the activity of endotype xanthanase was increased by 16 times after being fused with *Psp*CBM84. In addition, the presence of both CBMs obviously enabled endotype xanthanase to produce more oligoxanthan, and xanthan digests prepared by *Mi*Xen-CBM84 showed better antioxidant activity due to the higher content of active oligosaccharides. The results of this work lay a foundation for the rational design of endotype xanthanase and the industrial production of oligoxanthan in the future.

## 1. Introduction

Xanthan is an exopolysaccharide produced by the bacterium *Xanthamonas campestris* [1,2,3]. The cellulosic backbone of xanthan is linked by β-1,4-glycosidic bonds, and the β-D-Manp-(1→4) β-D-GlcUAp (GlcA)-(1→2) α-D-Man sidechain is bond with the glucose unit of backbone through the α-1,3-link (Figure 1) [4,5]. The terminal and the inner mannoses in the sidechain are variably pyruvylated or acetylated, and the frequency of the occurrence depends on particular *Xanthomonas* strains and fermentation conditions [6,7,8]. To date, xanthan hydrolysate with low molecular weight (LMW) have attracted extensive attention because of their excellent antioxidant activity, bacteriostatic activity and antitumor function, etc. [9,10]. While xanthan biodegradation is considered to be a promising strategy for the preparation of LMW products, the low enzymolysis efficiency of xanthan hinders the industrial production of functional LMW xanthan products.

As a potential tool for xanthan biodegradation, the endotype xanthanases belongs to the glycoside hydrolase and could specifically act on xanthan backbone (Figure 1) [5,11,12]. At present, only two endotype xanthanases have been reported, including *Psp*Xan9, a family 9 glycoside hydrolase (GH9) from *Paenibacillus* sp. 62047 [13], and *Mi*Xen, also a GH9 from *Microbacterium* sp. XT11 [14]. The molecular weights of the two enzymes are very similar, and both enzymes could randomly cleave glycosidic bonds within xanthan substrates, but their efficiency of cutting xanthan is significantly different [13,14]. Although *Psp*Xan9 can degrade xanthan more effectively, the highly ordered structure leads to the poor enzymatic accessibility of xanthan, which subsequently reduces the hydrolysis efficiency of endotype xanthanses [15].

So far, carbohydrate binding modules (CBMs) have been found in many CAZymes and proved that they can specifically recognize and bind to carbohydrate substrates [17]. The known CBMs are divided into 88 families and have broad affinities for a variety of carbohydrates such as cellulose, xylan, mannan, starch and chitin [18,19,20,21,22]. Based on structure and substrate affinity foundations [23], many cellulases have been engineered by fusing the C-terminus of the catalytic domain with a certain CBM, generating the recombinants with improved affinity and enzymolysis efficiency on cellulosic substrates [24,25,26]. For example, by fusing with the CBM from *Pseudomonas fluorescens*, the catalytic efficiency of the cellulase EGE has been increased by 2–6 times [27]; besides, after fusing with the CBM4-2, the recombinant enzyme AXE1dC-CBM4-2 has twice as much catalytic efficiency as AXE1dC [28].

Previous three-dimensional structural analysis [29,30] demonstrated that endotype xanthanase is a complex multimodular enzyme which is mainly composed of a catalytic domain, an *N*-terminal “immunoglobulin-like” domain and a *C*-terminal domain composed of β-sheets [14]. The *C*-terminal domain has been shown to function as a CBM, which helps these enzymes target the substrate xanthan [14]. So far, *Microbacterium* sp. XT11 (*Mi*) CBMx [14] and *Paenibacillus* sp. 62047 (*Psp*) CBM84 [13], from the endotype xanthanases *Mi*Xen and *Psp*Xan9, respectively, are the only CBMs that have been shown to be able to identify xanthan backbones. It is believed that rational engineering of the CBMs in xanthanases could improve the access of the enzymes to substrate and their efficiency of xanthan hydrolysis. However, a lack of in-depth investigation of the precise roles of CBMs in xanthan recognition and catalysis hinders such engineering processes. To address this issue, *Psp*CBM84 and *Mi*CBMx, derived from two different endotype xanthanases, were, respectively, fused with the catalytic domain of *Mi*Xen from *Microbacterium* sp. XT11. The effects of the two CBMs on the catalytic properties of *Mi*Xen were analyzed and compared. Our work lays a foundation for the rational engineering of endotype xanthanases.

## 2. Results and Discussion

### 2.1. Sequence and Structure Comparison of PspCBM84 and MiCBMx

To investigate the effects of CBMs on the catalytic properties of endotype xanthanase, two xanthanase-derived CBMs, *Psp*CBM84 and *Mi*CBMx were selected and compared. A crystal structure of *Psp*CBM84 (6fhj.1.A) is available (Figure 2a), and the amino acid sequence of *Mi*CBMx was submitted to SWISS-MODEL (https://swissmodel.expasy.org/interactive, accessed on 17 February 2023) for structural prediction. Only two prediction structures were given here, the sequence similarity between *Mi*CBMx and the templates, *Psp*CBM84 (6fjh.1.A) from *Paenibacillus* sp. 62047, and Lprg (3mh8.2.A) from *Mycobacterium Tuberosis* are 40.82% and 10.81%, respectively. Therefore, the *Mi*CBMx structure predicted with *Psp*CBM84 as the template was selected for subsequent analysis (Figure 2b). Further compare the structure of two CBMs, both CBMs contain a large amount of random coil structure in addition to the common alpha helix, beta turn and extended strand structures (Table 1). It is noteworthy that the percentage of alpha helical structure in the overall structure of *Psp*CBM84 (11.35%) is significantly higher than that in *Mi*CBMx (6.71%), which may lead to differences in their performance. The predicted three-dimensional structure of *Mi*CBMx can be aligned with that of *Psp*CBM84 with Cα root–mean–square deviation (RMSD) of 0.073 Å over 130 out of 149 amino acids, indicating a very high structural similarity between the two CBMs (Figure 2c) [31]. This similar structure indicates that the two CBMs may have similar functions in identifying xanthan substrates.

### 2.2. Construction and Characterization of Different Recombinant Forms of MiXen

To compare the functions of *Psp*CBM84 and *Mi*CBMx, three recombinant proteins- the catalytic domain of endotype xanthanase *Mi*Xen (*Mi*Xen-CD), *Mi*Xen-CBMx (containing *Mi*CBMx at the C-terminus), and *Mi*Xen-CBM84 (*Psp*CBM84 fused at the C-terminus of the catalytic domain of *Mi*Xen)-were constructed in this work or in previous work (Figure 3a) [14]. The recombinant proteins obtained were >90% pure and migrated as single bands with molecular weights of approximately 100 kDa (*Mi*Xen-CBMx), 80 kDa (*Mi*Xen-CD) and 92 kDa (*Mi*Xen-CBM84), respectively, in SDS-PAGE (Figure 3b).

The optimum reaction conditions and the stabilities of the three recombinant endotype xanthanases were measured when the substrate was xanthan. As shown in Figure 4a, the optimal temperature for the activity of *Mi*Xen-CBM84 was 55 °C, which was significantly lower than for *Mi*Xen-CD (65 °C) but higher than for *Mi*Xen-CBMx (40 °C) [14]. As shown in Figure 4b, *Mi*Xen-CBM84 retained >60% of its maximum activity after incubation for 1 h at 40 to 60 °C; *Mi*Xen-CBMx retained >60% of its maximum activity after incubation at 20 to 45 °C, and *Mi*Xen-CD retained >60% of its activity after incubation at 40 to 50 °C [14]. The high thermostability of *Mi*Xen-CBM84 is probably due to the thermotolerance of the *Psp*CBM84 domain, which is derived from the thermophilic bacterium *Paenibacillus* sp. 62047 [13]. Similar investigation has previously reported that the thermal stability of a pullulanase was significantly improved by fusion with a CBM from thermophilic strain *Anoxybacillus* sp. LM18-11 [32]. In addition, *Mi*Xen-CBMx shows stability at a low temperature. However, when the temperature is higher than 45 °C, its stability will decline rapidly, which proves that *Mi*CBMx may not work under high temperature [14]. The optimal pHs for reaction of *Mi*Xen-CBM84 (7.5) and *Mi*Xen-CBMx (7.0) [14] were lower than that of *Mi*Xen-CD (8.0), and the optimum pHs of all the three enzymes are shown in NaH_2_PO_4_-Na_2_HPO_4_ buffer (Figure 4c). *Mi*Xen-CBM84 retained >60% of its activity after incubation for 1 h at pH 6–9, *Mi*Xen-CBMx retained >60% of its activity at pH 5–9, and *Mi*Xen-CD retained >60% of its activity after incubation at pH 7–8.5 (Figure 4d). Notably, *Mi*Xen-CBMx retained the most enzymatic activity when placed in acidic (pH 4–6.5) environments, and *Mi*Xen-CBM84 retained the most enzymatic activity when placed in alkaline (pH 8–9.5) environments, these results may be caused by the different tolerance of the two CBMs to pH, which is also consistent with Zeng et al. [32]. In addition, when the pH of the buffer is higher than 9, the stability of all the enzymes will decline rapidly. Thus, both *Mi*Xen-CBM84 and *Mi*Xen-CBMx were stable at a wider pH range than *Mi*Xen-CD. These results confirmed that fusion with either CBM improved the pH stability of the endotype xanthanase.

The effects of 10 common different metal ions on the activities of the three endotype xanthanases were examined. As shown in Table 2, except for Ni^2+^ and Cu^2+^, which repressed the activity of all the enzymes, the effects of the other ions on the activities of the enzymes varied between the enzymes. For example, the presence of K^+^ repressed the activity of *Mi*Xen-CD, did not change the activity of *Mi*Xen-CBM84, but increased the activity of *Mi*Xen-CBMx. This investigation suggests that xanthanases with different CBM domains might recognize and bind different metal ions which disrupt or stabilize the conformation of the enzyme and lead to changes in the catalytic activity [33]. All the results revealed that basic properties of the endotype xanthanase *Mi*Xen-CD were affected to varying degrees by fusion with *Psp*CBM84 or *Mi*CBMx. This was similar to the result of Liu et al., who showed that the same metal ions (K^+^, Ca^2+^, Zn^2+^ et al.) have different effects on the activity of a novel *Aspergillus sulphureus* JCM01963 xylanase AS-xyn10A and its CBM-truncated variant (AS-xyn10A-dC) [18].

### 2.3. Xanthanase Fused with PspCBM84 Exhibits Better Hydrolysis Performance against Xanthan

It has been shown that, by linking specific CBMs to the catalytic domain, the catalytic efficiency of an enzyme can be improved [34,35]. As shown in Table 3, the apparent *K*_m_ values of the recombinant xanthanases fused with CBMs here were significantly lower than that of *Mi*Xen-CD, suggesting high affinities of both *Psp*CBM84 and *Mi*CBMx toward xanthan. In addition, due to the similarity of the main chain structure of xanthan and cellulose, the *K*_m_ value of some endocellulase can be used to evaluate the endotype xanthanases, such as the cellulases from *Aspergillus niger* [36], *Thermomyces duponti* [37] and *Bacillus* sp., respectively [38], the *K*_m_ of which are about 0.23 g/L–0.8 g/L, which further proves the feasibility of our kinetic model. The results were also in consistent with surface plasmon resonance (SPR) analysis (Figure 5). Both *Mi*Xen-CBM84 and *Mi*Xen-CBMx showed concentration-dependent binding of xanthan, and the dissociation constant (*K*_D_) values for xanthan from *Mi*Xen-CBM84 and *Mi*Xen-CBMx were lower than from *Mi*Xen-CD. *k*_cat_/*K*_m_ values indicated that the catalytic efficiency toward xanthan was dramatically improved by fusion of the CBMs to *Mi*Xen. All these results reveal that both CBMs play an important role in xanthan enzymolysis by improving the access of the enzyme to the xanthan substrate [28,32,39]. Notably, the highest substrate affinity, catalytic efficiency, and specific activity were observed for *Mi*Xen-CBM84, which indicated that *Psp*CBM84 might endow the xanthanase with excellent performance in xanthan hydrolysis (Table 2). Although the activity of *Mi*Xen-CBM84 (82.4 ± 5.18 U/g) was not as high as that of *Psp*Xan9 (285 ± 10 U/g) [13], the addition of *Psp*CBM84 increased the activity of *Mi*Xen (5.1 ± 0.59 U/g) 16-fold (Table 3). The difference in activity between *Mi*Xen-CBM84 and *Psp*Xan9 may result from their different catalytic domains.

Atomic force microscopy (AFM) was employed to morphologically investigate xanthan treated by the different recombinant xanthanases. As shown in Figure 6, the untreated xanthan sample had a tree-like structure with many sharp kinks, and its average height was estimated to be 0.64 ± 0.07 nm (Figure 7d). After incubation with *Mi*Xen-CD, *Mi*Xen-CBMx, and *Mi*Xen-CBM84, respectively, for 12 h, the average height of the xanthan samples decreased to 0.54 ± 0.06, 0.48 ± 0.08, and 0.40 ± 0.06 nm, respectively (Figure 7d). Moreover, more random-coil structures and short chains were observed in the xanthan samples. The results showed that the original structure of xanthan was degraded by the enzymes, and a more disordered state was observed after treatment with them, which is consistent with previous reports [14]. Both CBMs can help *Mi*Xen better open the xanthan structure and increase the binding degree of *Mi*Xen with the substrate and cut the main chain of xanthan more effectively. The lowest mean height was observed when xanthan was treated with *Mi*Xen-CBM84, suggesting the highest hydrolytic efficiency was obtained when using this enzyme, consistent with the measured kinetic properties of the enzymes (Table 3).

The role of the CBM domains in enzymolysis of xanthan was verified using gel permeation chromatography (GPC), high-performance liquid chromatography (HPLC) and ultra-high performance liquid chromatography/quadrupole time-of-flight mass spectrometry (UPLC-QTOF-MS). Firstly, the molecular weight distributions of xanthan treated by the different endotype xanthanases were detected by GPC. As shown in Figure 7a, the original xanthan had a high molecular weight (3 to 4 × 10^6^ Da) and a retention time (Rt) of 18–21 min. After incubation with the respective endotype xanthanases, in addition to residual xanthan (Rt = 18–21 min) that was not degraded, some products with intermediate molecular weight (6 to 9 × 10^5^ Da) (Rt = 21–24 min) appeared in the samples. *Mi*Xen-CBM84 (49.6%) converted more xanthan into intermediate-molecular-weight products than *Mi*Xen-CD (25.6%) or *Mi*Xen-CBMx (29.8%) (Table 4), indicating that *Psp*CBM84 could better assist the endotype xanthanase to bind and cleave the xanthan backbone.

HPLC analysis of the xanthan digests produced by the three endotype xanthanases indicated that the enzyme forms with fused CBM domain could release more oligoxanthan products from xanthan than did *Mi*Xen-CD. The highest content of oligoxanthan was obtained when xanthan was hydrolyzed by *Mi*Xen-CBM84 (Figure 7b), which is consistent with the observation from GPC. To determine the composition of the oligoxanthan observed in HPLC analysis, the same digested samples were analyzed using UPLC-QTOF-MS. The UPLC-QTOF-MS analysis revealed that the major oligoxanthan products produced by *Mi*Xen-CBM84 were an unsaturated tetrasaccharide (661 *m/z*, Figure 8a) and monoacetylated derivatives (703 *m/z*, Figure 8b). This was similar to the result of Moroz et al., who showed that endotype xanthanase (*Psp*Xan9) can generating tetrameric xanthan oligosaccharides from polysaccharide lyase family 8 (PL8) xanthan lyase-treated xanthan [13]. The abundances of the constituents in the three xanthan hydrolysates were different (Figure 8c,d). However, the presence of tetrasaccharide products demonstrated that all the recombinant enzymes are capable of cleaving lyase-treated xanthan at intervals of two glucose residues in the backbone. All the above investigations established that the CBM domain plays an important role in xanthan enzymolysis.

So far, many cellulases [23,26] and chitinases [19,22] have been engineered by fusing the C-terminal of the catalytic domain with a certain CBM, generating the recombinant enzymes with improved affinity and enzymolysis efficiency on substrates; however, there is no report on the application of CBMs in endotype xanthanases modification. In this work, by fusing two CBMs (*Psp*CBM84 and *Mi*CBMx) with endotype xanthase (*Mi*Xen-CD), respectively, we proved for the first time that the feasibility of CBMs in the rational design of endotype xanthanase. Moreover, the high catalytic efficiency of *Mi*Xen-CBM84 proves that *Psp*CBM84 is a better module.

### 2.4. Xanthan Digests Prepared by MiXen-CBM84 Showed Improved Antioxidant Activity

The antioxidant activities of xanthan digests produced by the three recombinant endotype xanthanases were evaluated. The half-inhibitory concentration (IC_50_) values of the products prepared using *Mi*Xen-CBM84 in assays of ferrous ion chelating activity (Figure 9a), scavenging activity of the DPPH (1,1-diphenyl-2-picrylhydrazyl) radical (Figure 9b), and hydroxyl radical scavenging activity (Figure 9c) were lower than those of the products prepared using *Mi*Xen-CBMx and *Mi*Xen-CD treated samples. The highest reducing power was also observed for hydrolysate prepared using *Mi*Xen-CBM84 (Figure 9d). The antioxidant capacity of the hydrolysates obtained by the enzymatic method was greater than of those obtained by acid or alkaline oxidation methods [9]. The xanthan hydrolysates produced by *Mi*Xen-CBM84 possess excellent antioxidation performance. It has been confirmed previously that the different proportions of active components in the hydrolysates of polysaccharides lead to differences in their antioxidant properties [41]. The DE valve was used to calculate the yield, compared with *Mi*Xen-CD (0.15%) and *Mi*Xen-CBMx (0.46%), the reaction of *Mi*Xen-CBM84 (2.39%) with xanthan produced a higher yield of oligoxanthan (Table 3).

It has been reported that the antioxidant performance of xanthan hydrolysates is affected by their pyruvate group content [9,42]. Therefore, we characterized the xanthan hydrolysates produced in this study by infrared spectroscopy. The absorption peak at 1600 cm^−1^ (Figure 7c) represents the C=O double bond in the pyruvic acid group. The product obtained from the hydrolysis of xanthan by *Mi*Xen-CBM84 had the most obvious absorption peak at 1600 cm^−1^, consistent with the xanthan digests produced by *Mi*Xen-CBM84 having the highest antioxidant activity. In addition, by comparing the hydrolysis efficiency of endotype xanthanase and the relative content and antioxidant activity of its xanthan digests, we believed that during the degradation of xanthan, the higher the catalytic efficiency of endotype xanthanase, the looser the structure of the xanthan digest, and the more pyruvate group in the original dense structure will be exposed, which further leads to higher antioxidant activity of its digests.

## 3. Materials and Methods

### 3.1. Sequence Analysis and Structural Alignment of Enzymes

Tools at Prabi (https://npsa-prabi.ibcp.fr/cgi-bin/npsa_automat.pl?page=npsa_sopma.html, accessed on 17 February 2023) were applied to predict and compare the secondary structure of *Psp*CBM84 and *Mi*CBMx. SWISS-MODEL (https://swissmodel.expasy.org/interactive) was used to predict the three-dimensional structure of *Mi*CBMx. PyMOL visualization software was applied to compare the features between the model of *Mi*CBMx with the crystal structure of *Psp*CBM84 (6fhj.1.A).

### 3.2. Strains, Plasmids and Culture Conditions

In previous work, a xanthan-degrading strain named *Microbacterium* sp. XT11 (China Center for Type Culture Collection CCTCC AB2016011) was isolated in the author’s laboratory [14]. In this work, the gene encoding the endotype xanthanase *Mi*Xen, and the gene fragment encoding its catalytic domain (*Mi*Xen-CD), were cloned from genomic DNA of *Microbacterium* sp. XT11. *Escherichia coli* DH10B and *E. coli* BL21 (DE3) were used in gene cloning experiments and for the expression of recombinant enzymes, respectively.

*Microbacterium* sp. XT11 was cultivated at 30 °C in xanthan medium (0.5 g/L glucose, 3 g/L yeast extract, 3 g/L xanthan (purchased from sigma), 0.8 g/L NaCl, 0.05 g/L K_2_HPO_4_, 0.70 g/L KNO_3_ and 0.025 g/L MgSO_4_·7H_2_O, pH 7.0). *E. coli* strains were grown at 37 °C in Luria–Bertani (LB) medium (5.0 g/L yeast extract, 10 g/L sodium chloride and 10 g/L tryptone, pH 7.0) [14].

### 3.3. Cloning, Expression and Purification of Endotype Xanthanase Constructs

The gene encoding *Mi*Xen-CD was amplified by PCR from genomic DNA from *Microbacterium* sp. XT11, then inserted into vector pET28a by restriction-free cloning [43]. The forward primer was 5′-GCAAATGGGTCGCGGATCCGAATTCATGTCCCGACGACGAGCGAG-3′, and the reverse primer was 5′-CACCACCACCACCACCACCGGCATGGTTGTTGTACCAATTCGCC-3′. To construct the mutant *Mi*Xen-CBM84, the genes encoding *Mi*Xen-CD and *Psp*CBM84 were amplified by PCR using genomic DNA from *Microbacterium* sp. XT11 and plasmid pUC57-CBM84, synthesized by Shanghai Shenggong Biotechnology Co., Ltd., as the templates, respectively. Then, the sequence encoding *Psp*CBM84 was fused at the 3′-terminus of the sequence encoding *Mi*Xen-CD by overlapping extension PCR [44]. For cloning *Mi*Xen-CD, the forward primer was 5′-GCGAATTCATGTCCCGACGACGA-3′, and the reverse primer was 5′-GTGTAAACGCGGCCGTCAGAACCACCACCACCACCGGCAT-3′. For cloning the *Psp*CBM84, the forward primer was 5′-ATGCCGGTGGTGGTGGTGGTTCTGACGGCCGCGTTTACAC-3′, and the reverse primer was 5′-GCAAGCTTTCAGTTACCCAGCGCGT-3′.

The PCR products were digested with *Eco*RI and *Hin*dIII, then ligated into *Eco*RI-and *Hin*dIII-cut pET28a. A 6×His tag-encoding sequence was added to the 3′-terminus of DNA products to facilitate downstream protein purification. Next, by using a Gene Pulser, the correct plasmids, pET-*Mi*Xen-CD and pET-*Mi*Xen-CBM84 were, respectively, transformed into electrocompetent *E. coli* BL21 (DE3). The electroporation conditions were 2.5 kV, 25 μF, and 200 Ω. The recombinants strains were cultivated at 37 °C on LB-agar plates containing 30 mg/L kanamycin, and then screened by colony PCR.

For the expression of the endotype xanthanases *Mi*Xen-CBMx, *Mi*Xen-CD, and *Mi*Xen-CBM84, transformants were cultivated at 37 °C in LB medium containing 30 mg/L kanamycin until OD_600nm_ reached 0.5; then, isopropyl β-D-1-thiogalactopyranoside was added (final concentration: 0.5 mM), and the transformants were cultivated at 16 °C for a further 16 h. Next, cells were collected and fragmented by centrifugation and ultrasonic processing, respectively. After collecting the supernatant, protein was purified using a Ni-NTA chromatography purification system [45]. Finally, the purity of proteins was assessed by SDS-PAGE, and the concentration of purified protein was determined by using a BioRad Protein Assay Kit (BioRad, Hercules, CA, USA).

### 3.4. Enzyme Activity Assays

Xanthan solution (Sigma) with concentration 2 g/L was used for determination of enzyme activity. The final concentration of the three endotype xanthanases used in all assays was 0.2 mg/mL (1.3 times that used by Yang) [14]. The reaction was conducted at 40 °C for 20 min, then the reaction mixture was boiled for 10 min to terminate the reaction. The amount of reduced sugar released from the reaction was used as an index to evaluate the reaction and was measured by the bicinchoninic acid method. One unit of enzyme activity was defined as the amount of enzyme required to produce 1 μmol reducing sugar per min.

### 3.5. Biochemical Characteristics of the Recombinant Endotype Xanthanases

To measure the optimum reaction temperatures of the enzymes, the enzyme reaction system was placed at different temperatures (20 to 70 °C). In thermal stability experiments, the enzyme was preincubated at 20–70 °C for 1 h and the residual activity was measured using the standard reducing end assay as described above at the temperatures where the respective maximal enzyme activities were observed [46]. To measure the optimal pHs for the enzyme activities, we used various buffer systems (final concentrations: 10 mM): HOAc-NaOAc buffer, pH 4.0 to 6.0; NaH_2_PO_4_-Na_2_HPO_4_ buffer, pH 6.0 to 8.0; Tris-HCl buffer, pH 8.0 to 9.0; and NaHCO_3_-NaOH buffer, pH 9.5 to 10.0. The reactions were performed at 40 °C. In pH stability experiments, the enzyme was preincubated at pH 4.0–10.0 for 1 h and the residual activity was measured using the standard reducing end assay as described above at the pH where the maximal activity for the respective enzyme was observed [46]. The temperature/pH range where the residual enzyme activity remained above 60% of the maximum activity was defined as the temperature/pH stability range of the enzyme [47]. After identifying the optimum reaction conditions of the various enzymes, a series of xanthan solutions (0.1, 0.25, 0.5, 0.75, 1.0, 1.5, 2.0, 2.5, 5.0 and 10.0 g/L) was prepared and used for determination of the kinetic parameters (*K*_m_ and *V*_max_) of the enzymes. The initial velocity of the reaction was determined, and then the Michaelis–Menten equation was used to calculate apparent *K*_m_ and *V*_max_ values.

To verify the effects of 10 types of common metal ion on the activity of the three endotype xanthanases, the ions were separately added to xanthan solutions at a final concentration of 1 mmol/L [14,15,16,47,48]. Then, the enzyme activity was determined by the standard method. The reaction between xanthan and enzyme solution in the absence of added metal ions was used as the control, for which the activity was defined as 100%.

### 3.6. Surface Plasmon Resonance

Surface plasmon resonance (SPR) measurement was performed on a BIAcore T200 instrument with a sensor chip with dextran matrix and pre-coated streptavidin (GE Healthcare). Xanthan with various concentrations (0.01, 0.1, 0.6, 4, and 15 nM) were prepared as analysis samples. SPR measurement was performed using a BIAcore T200 instrument with a sensor chip with dextran matrix precoated with streptavidin (GE Healthcare). Xanthan at various concentrations (0.01, 0.1, 0.6, 4, and 15 nM) was prepared as the samples for analysis. The chip was activated with 420-s injections of 0.05 M N-hydroxysuccinimide and 0.2 M ethyl-3-(3-dimethylaminopropyl) carbodiimide hydrochloride. Then, enzyme in running buffer (10 mM NaH_2_PO_4_-Na_2_HPO_4_, pH 7.5) was injected and captured on the channel surface, to achieve an immobilization level of about 2000 response units. For substrate binding experiments, the temperature of the instrument was set to 25 ± 0.01 °C, and the flow rate was set to 30 µL/min. Then, the sample solutions were injected onto the chip surface for 120 s. After the binding reaction, a dissociation time of 600–3000 s was applied, and, as regeneration buffer, 5 M NaCl solution was injected for 60 s to allow the signal to return to baseline.

### 3.7. Gel Permeation Chromatography

By using an Agilent 1260 infinity system (Agilent Technologies, Santa Clara, CA, USA), the molecular mass of xanthan digestion products was determined. The system had three in-series gel columns, including PL aquagel-OH 60, PL aquagel-OH MIXED-M, and PL aquagel-OH columns. The column temperature was set to 40 °C. NaNO_3_ (0.1 M) was used to dissolve and elute the samples with a flow rate of 0.8 mL/min. The eluate was monitored using a refractive index detector. Pullulan molecular-mass standards (Polymer Laboratories, Palo Alto, CA, USA) were used to estimate molecular weights.

### 3.8. AFM Imaging of Xanthan Samples

To allow imaging of individual well-separated xanthan molecules, 5 μL aliquots of undigested or digested xanthan (10 mg/L) were pipetted onto freshly cleaved mica sheets (Park Systems, Suwon, Republic of Korea) and dried under a gentle flow of dry nitrogen gas. Topographical and error signal mode imaging of the samples was then conducted using a commercial atomic force microscope (XE-Bio, Park Systems, Republic of Korea) in non-contact mode with NCHR monolithic silicon cantilevers (Park Systems, Republic of Korea). During the measurements, an isolation platform and an acoustic enclosure were used together to reduce environmental interference with the experiment. The samples were scanned using the AFM controller software xep (version 1.7.70.3); the scanning rate was 0.5 Hz, and the resolution of the scanned images was 256 × 256 pixels. The height of individual xanthan strands was estimated by XEI. The height of an individual strand was evaluated by averaging the height estimated from at least five cross sections. To estimate the mean height, the overall mean of one sample was calculated.

### 3.9. Liquid Chromatography Analysis

The size distribution of the products of digestion of xanthan treated by xanthan lyase [49] and endotype xanthanases was analyzed by HPLC (1260 Infinity II system, Agilent Technology, Palo Alto, CA, USA). The chromatographic separation column used a 4.6 mm × 250 mm amide column with working temperature 35 °C. A mixture of acetonitrile (50%), water (40%), and ammonium formate (10%) was used as the mobile phase; 10 μL samples were injected each time for analysis with flow rate 1.0 mL/min. An evaporative light scattering detector was employed. The elution mode was gradient elution, and the elution time was 30 min.

### 3.10. UPLC-QTOF-MS Analysis

The composition of the products of digestion of xanthan was determined by UPLC-QTOF-MS using an Agilent 6545 instrument in negative ion mode with integration time 2 s. The operating voltages of the fragment, skimmer, and octopole RF were set to 75, 60, and 750 V, respectively. The temperature and flow rate of the dry gas were 350 °C and 7.2 L/min, respectively. The MS scanning range was *m/z* 100–1700. A mixture of water and 50% acetonitrile was used as the mobile phase, with flow rate 0.2 mL/min; 1 μL samples were injected for analysis.

### 3.11. Antioxidant Activity Analysis of Xanthan Digests

A 10 mL reaction was carried out by the standard enzyme reaction process. The yield was calculated as the DE value; the DE value represents glucose equivalents, which means that the percentage of reducing sugar in dry matter was calculated by taking all the reducing sugar in the reaction system as glucose [40]. The hydrolysate sample used was the dry powder obtained after removal of the enzyme, concentration by ultrafiltration, and freeze-drying. An equal mass of hydrolysate was used in each antioxidant activity test. Four indicators were used in antioxidant activity assays, including DPPH (1,1-diphenyl-2-picrylhydrazyl) radical scavenging activity, hydroxyl radical scavenging activity, reduction force and the chelating activity of ferrous ions. The DPPH radical scavenging activity was determined by Yamaguchi’s method [50]. Measurement of hydroxyl radical scavenging activity was based on the Fenton reaction [51]. The reduction force was determined by the method of Yen and Chen [52]. The chelating activity of ferrous ions was determined by the method of Chan [53].

### 3.12. Infrared Spectroscopy

Fourier transform infrared spectroscopy (Nicolet iS5 FTIR ThermoFisher) was used to analyze digested xanthan. The test mode was set to through, and the resolution of the mid-infrared zone and the near-infrared resolution were set to better than 0.4 cm^−1^ and 0.1–6.4 nm, respectively. The spectral range was 15,000–450 cm^−1^, and the signal-to-noise ratio was set to lower than 250,000:1. The scanning speed was 0.1–4.0 cm/s, and the wavenumber accuracy was set to better than 0.008 cm^−1^.

### 3.13. Statistical Analysis

Three parallel samples were measured each time in all of the assays and analyses. Values are exhibited as the mean ± standard error. SPSS 11 (SPSS Inc., Chicago, IL, USA) was used to evaluate the significant difference at *p* < 0.05.

## 4. Conclusions

In conclusion, the function of two different CBMs on catalytic properties of endotype xanthanase towards xanthan were investigated and compared in this work. The results revealed that both *Psp*CBM84 and *Mi*CBMx could change the basic characterizations of the xanthanase and could endow the enzyme with better catalytic properties through enhancing the accessibility of the xanthan substrate to enzyme. Notably, the recombinant enzyme fused with *Psp*CBM84 exhibited excellent xanthan enzymolysis performance, and the obtained xanthan hydrolysates possessed excellent antioxidation property. Our work should be valuable for the rational engineering of xanthanases and the production of functional oligosaccharides.

## Figures and Tables

**Figure 1 ijms-24-05480-f001:**
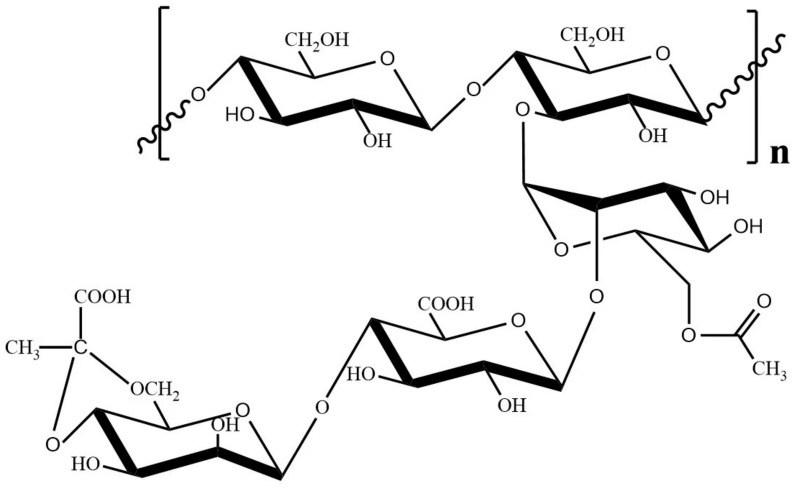
The structure of xanthan [16].

**Figure 2 ijms-24-05480-f002:**
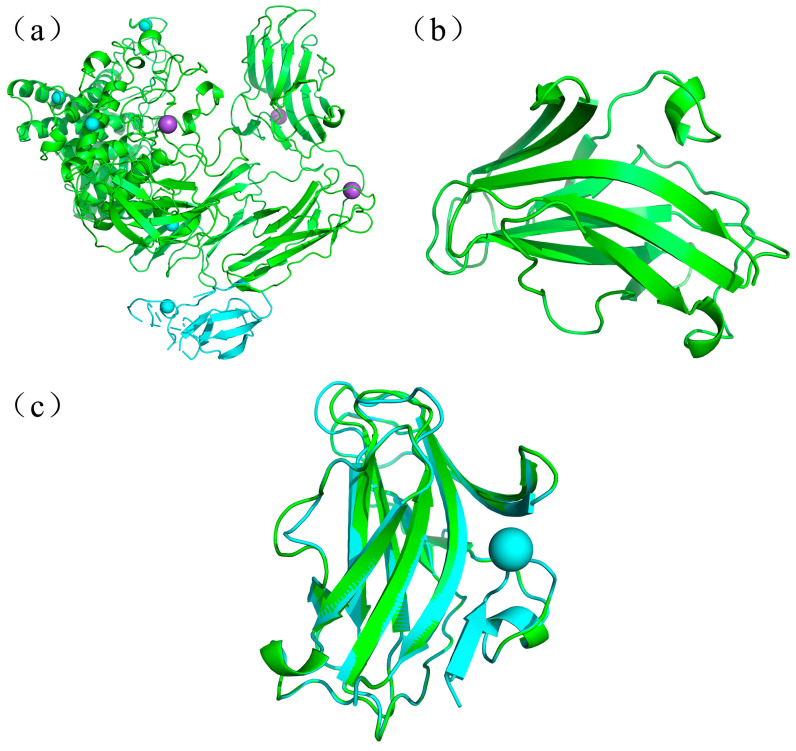
Structure properties of *Mi*CBMx and *Psp*CBM84. (**a**) The structure of *Psp*Xan9 (6fhj), in which the blue part is the structure of *Psp*CBM84. (**b**) The prediction three dimensional structure of *Mi*CBMx. (**c**) Three-dimensional structure comparison of the *Psp*CBM84 (blue) and *Mi*CBMx (green). The calcium ions are shown as cyan spheres, sodium as purple [13].

**Figure 3 ijms-24-05480-f003:**
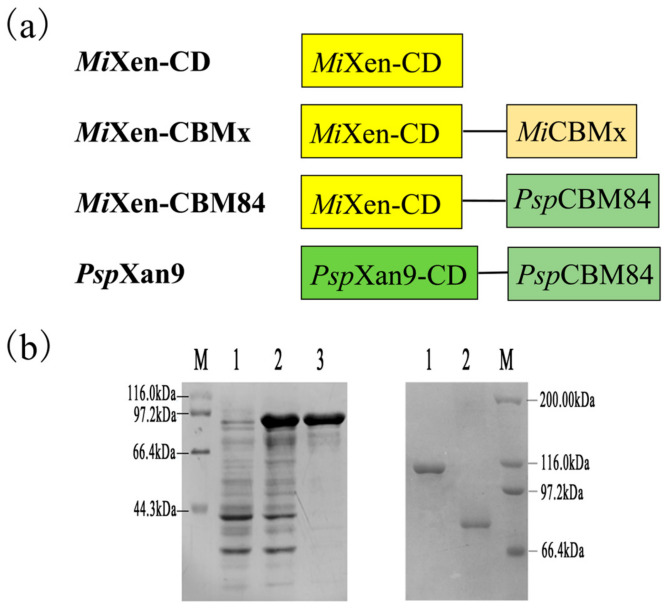
Rational design and SDS-PAGE analysis of recombinant enzymes used in this work. (**a**) Schematic structures of the endotype xanthanases used in this paper. (**b**) SDS-PAGE analysis of endotype xanthanases. Left: *Mi*Xen-CBM84. Lanes M: protein molecular weight markers, Lanes 1: carrier protein, Lanes 2: soluble protein, Lanes 3: purified protein. Right: SDS-PAGE analysis of *Mi*Xen-CBMx and *Mi*Xen-CD. Lanes M: protein molecular weight markers. Lanes 1: purified *Mi*Xen-CBMx. Lanes 2: purified *Mi*Xen-CD.

**Figure 4 ijms-24-05480-f004:**
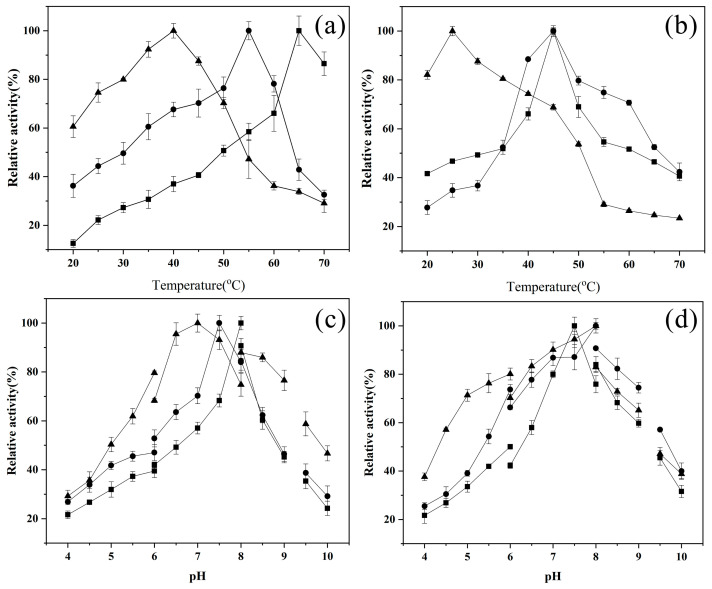
Biochemical characteristics of *Mi*Xen-CD (■), *Mi*Xen-CBM84 (●) and *Mi*Xen-CBMx (▲) [14]. (**a**) The optimun reaction temperatures of the endotype xanthanases with xanthan as the substrate. (**b**) The thermostability of the endotype xanthanases. (**c**) The optimum reaction pHs of the endotype xanthanases with xanthan as the substrate. (**d**)The pH stability of three endotype xanthanases. Buffers: HOAc-NaOAc buffer (pH 4.0 to 6.0), NaH_2_PO_4_-Na_2_HPO_4_ buffer (pH 6.0 to 8.0), Tris-HCl buffer (pH 8.0–9.0), and NaHCO_3_-NaOH buffer (pH 9.5 to 10.0).

**Figure 5 ijms-24-05480-f005:**
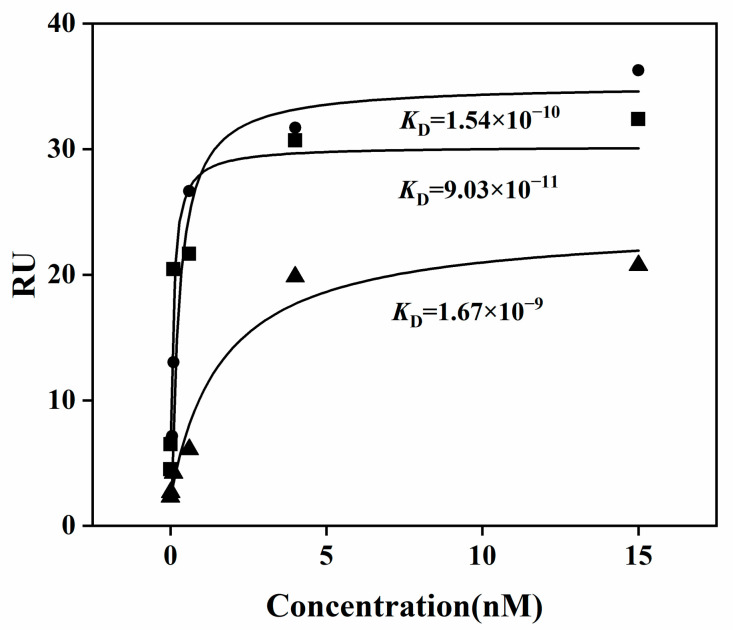
Surface plasmon resonance analysis of the interaction of the respective endotype xanthanases with xanthan. The dissociation constant (*K*_D_) values were calculated using a one-site steady state binding model.

**Figure 6 ijms-24-05480-f006:**
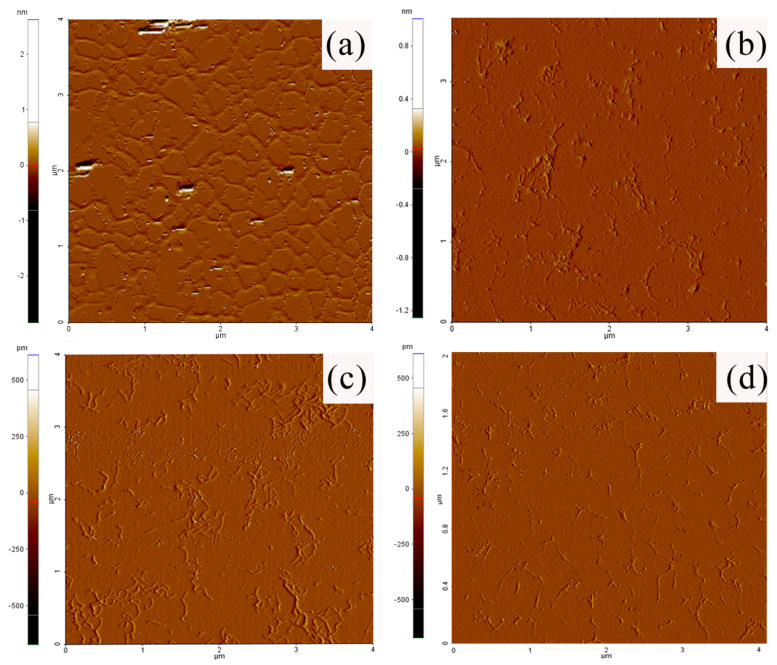
Atomic force microscopy (AFM) analysis of xanthan and xanthan digests. (**a**–**d**) AFM topography images of xanthan, xanthan hydrolyzed by *Mi*Xen–CD, xanthan hydrolyzed by *Mi*Xen–CBMx, and xanthan hydrolyzed by *Mi*Xen–CBM84, respectively.

**Figure 7 ijms-24-05480-f007:**
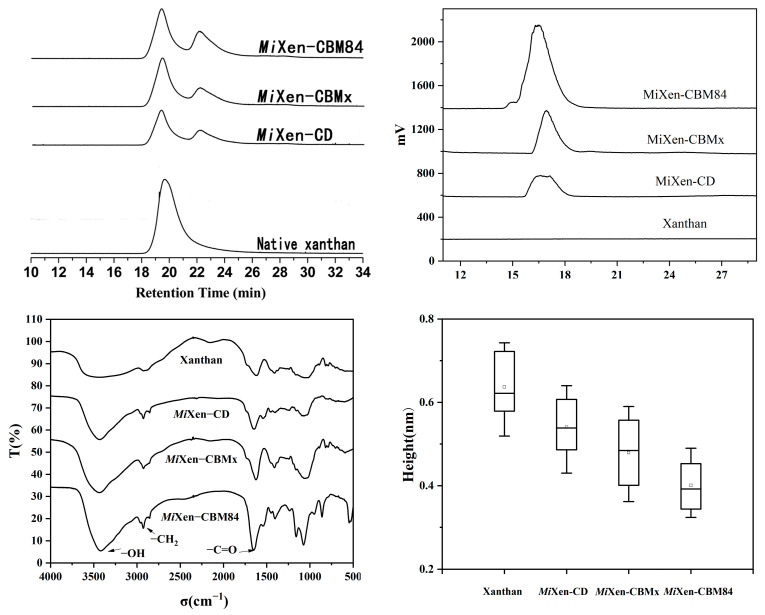
Analysis of xanthan hydrolysates. (**a**) Gel permeation chromatography elution patterns of xanthan and xanthan digests. (**b**) Liquid chromatography analysis of the degree of polymerization of hydrolyzed xanthan. (**c**) Infrared spectroscopic analysis of xanthan and its hydrolysates. (**d**) Boxplots of the measured heights of xanthan and xanthan hydrolysates (*n* = 20 for each).

**Figure 8 ijms-24-05480-f008:**
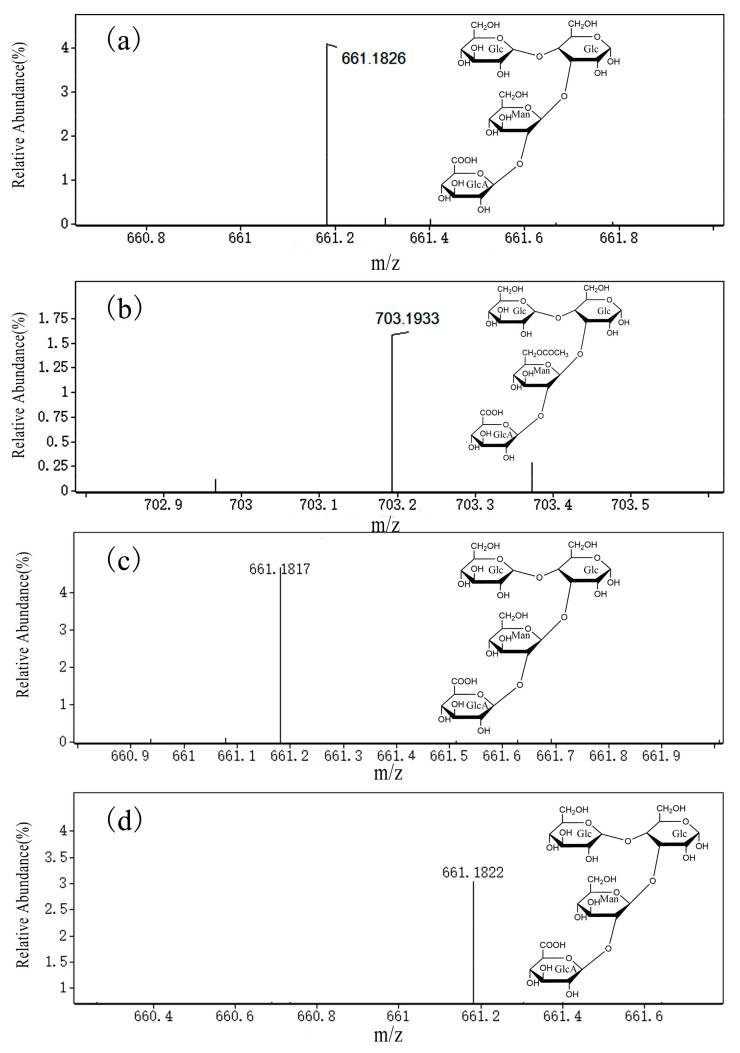
Ultra-high-performance liquid chromatography/quadrupole time-of-flight mass spectrometry analysis of the unsaturated tetrasaccharides produced by hydrolysis of xanthan. (**a**,**b**) Xanthan digested by *Mi*Xen-CBM84; (**c**) xanthan digested by *Mi*Xen-CBMx; (**d**) xanthan digested by *Mi*Xen-CD.

**Figure 9 ijms-24-05480-f009:**
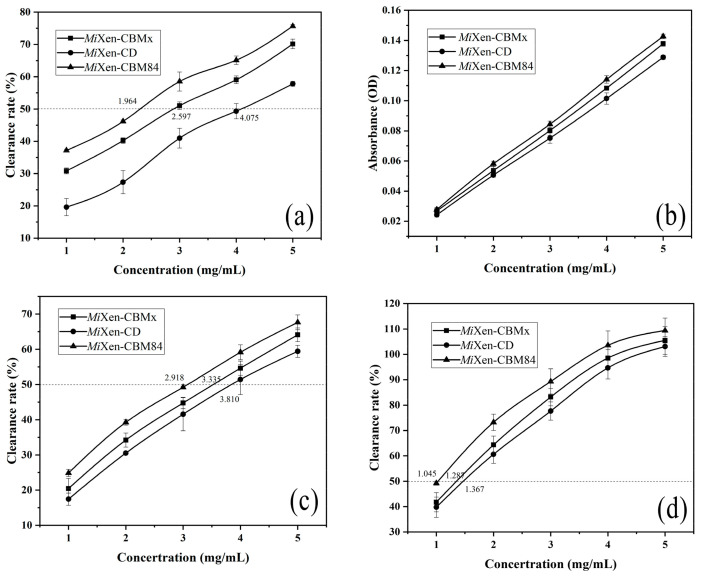
Evaluation of antioxidant properties of xanthan digests prepared by using *Mi*Xen-CD (●), *Mi*Xen–CBMx (■), or *Mi*Xen–CBM84 (▲). (**a**) Ferrous ion chelating activity of xanthan digests. (**b**) 1,1-Diphenyl-2-picrylhydrazyl (DPPH) radical scavenging activity of xanthan digests. (**c**) Hydroxyl radical scavenging activity of xanthan digests. (**d**) Reducing power of xanthan digests.

**Table 1 ijms-24-05480-t001:** Structural information of *Psp*CBM84 and *Mi*CBMx.

Enzyme	Alpha Helix	Extended Strand	Beta Turn	Random Coil
*Psp*CBM84	11.35%	34.04%	8.51%	46.1%
*Mi*CBMx	6.71%	34.90%	9.40%	48.99%

**Table 2 ijms-24-05480-t002:** Influence of different metal ions on activities of the endotype xanthanases.

Reagents Added	Relative Activity (*Mi*Xen-CD)	Relative Activity (*Mi*Xen-CBM84)	Relative Activity (*Mi*Xen-CBMx)
None	100	100	100
K^+^	70.7 ± 8.29	96.3 ± 2.29	114.1 ± 1.64
Fe^2+^	76.3 ± 5.45	126.8 ± 5.26	76.3 ± 6.86
Fe^3+^	123.8 ± 8.66	81.4 ± 3.45	87.8 ± 1.92
Ni^2+^	93.8 ± 3.59	91.1 ± 1.94	81.2 ± 2.86
Co^2+^	103.7 ± 5.69	138.7 ± 4.87	93.2 ± 0.99
Cu^2+^	53.3 ± 6.51	70.6 ± 3.60	64.9 ± 3.16
Zn^2+^	89.1 ± 4.84	89.2 ± 2.77	109.8 ± 3.22
Mn^2+^	44.5 ± 4.19	92.4 ± 4.77	120.4 ± 5.40
Mg^2+^	117.7 ± 4.70	116.2 ± 3.55	70.5 ± 2.15
NH_4_^+^	103.4 ± 5.83	67.8 ± 2.70	71.9 ± 1.98

The final concentration of the added ion was 1 mM. The activity in the enzyme reaction with xanthan as the substrate in the control group (i.e., with no added ion) was defined as 100%. The results for the experimental groups are expressed relative to the activity of the control group.

**Table 3 ijms-24-05480-t003:** Hydrolysis properties of the endotype xanthanases toward the xanthan backbone [13,14].

Enzyme	V_max_ (μmol/L/min)	*K*_m_ (g/L)	*k*_cat_/*K*_m_ (L/g·min)	Specific Activity (U/g)	DE Value (%)
*Mi*Xen-CD	(0.33 ± 0.05) ^a^	(1.18 ± 0.02) ^a^	(2.86 ± 0.05) ^a^	(5.1 ± 0.59) ^a^	(0.0092) ^a^
*Mi*Xen-CBM84	(6.57 ± 0.98) ^b^	(0.54 ± 0.03) ^b^	(121.7 ± 15.76) ^b^	(82.4 ± 5.18) ^b^	(0.1593) ^b^
*Mi*Xen-CBMx	(0.98 ± 0.02) ^c^	(0.65 ± 0.02) ^c^	(15.1 ± 0.38) ^c^	(15.7 ± 0.98) ^c^	(0.07) ^c^

Different superscripts in the same column indicate significant differences (*p* < 0.05), while the same superscripts in the same column indicate no significant differences (*p* > 0.05). DE value represents glucose equivalent, which means that the percentage of reducing sugar in dry matter is calculated by taking all reducing sugar as glucose in the reaction system [40].

**Table 4 ijms-24-05480-t004:** Quantitation of peak areas in gel permeation chromatography.

Enzyme	Xanthan	*Mi*Xen-CD	*Mi*Xen-CBM84	*Mi*Xen-CBMx
HMW	124.67	60.36	66.23	67.15
LMW	0	20.72	43.31	28.45

HMW: high-molecular-weight xanthan; LMW: intermediate-molecular-weight xanthan.

## Data Availability

All data produced or analyzed for this study are included in the published article and its additional information files.

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
