# Peer review of "An Insight into the Essential Role of Carbohydrate-Binding Modules in Enzymolysis of Xanthan"

_ijms, 2023, doi:10.3390/ijms24065480_

Round 1

Reviewer 1 Report

In the research article submitted by Xin Ni et al. the authors compare four recombinantly produced xanthanases (2 x native sequence, 1 x truncated without CBM and 1 x with exchanged CBM) and studied the effect of the CBM on the biochemical properties and activity. They could show that the chimeric enzyme (consisting of the catalytic domain of MiXen and the CBM of PspXan9) significantly increased the activity and stability of native MiXen and thereby the authors demonstrated the strong influence of the CBM towards the overall enzymatic efficiency.

Although the authors demonstrated the importance of the CBM domain and the methodology is interesting, they do not provide any novel insights into the mechanism of enzymatic xanthan degradation and they also could not exceed the activity of the previously characterized PspXan9. Furthermore, some details are in my opinion not clear and some interpretations are not very convincing. Therefore, I would suggest a thorough revision of the manuscript:

Major:

Structural analysis of CBMx and CBM84: The authors compare the previously published crystal structure of PspCBM84 with a I-Tasser model of MiCBMx. It would definitely be relevant to discuss the quality of the obtained model and not only the quality of the alignment. In my opinion, the predicted structure in Figure 1 b indicates a rather poor quality. Furthermore, the authors state: “The secondary structure comparison showed that both CBMs contain typical 4 turns, 2 η-helixes and β-sheets (Figure 1).” This is definitely not true from the data shown. In the predicted model of MiCBMx only 6 beta sheet elements are visible to me (Figure 1b). Furthermore, the one part of the sequence of PspCBM84 that contains 3 beta sheets does not at all align with the sequence of MiCBMx. In general, I believe the authors need to revise the section on the structural properties of the CBMs. I also believe that a structural alignment of the sequences (as presented in Figure 1a) for two sequences with only 20% identity and obviously such large structural differences is generally very problematic. Additionally, the term “η-helixes” is in my opinion very unusual and I think the authors mean 310 helix by that.

Biochemical characterization - Figure 3: The authors used 4 different buffer systems to measure their pH-dependent activity and stability. They should indicate this in the figure and the figure caption. Not only pH value but also buffer composition could influence activity as well as stability.

Line 129f: The authors write “As shown in Table 1, except for Ni2+, Cu2+ and NH4+, which were inhibitors for all the three recombinants, the other metal ions activated or repressed the enzyme activities in completely different ways.” How do the authors come to this conclusion and how do they differentiate between “inhibitors” and metal ions that “repressed” enzyme activity? This is not conclusive to me. Furthermore, as NH4+ (described as inhibitor by the authors) shows an activating or at least neutral effect on MiXen-CD (103.4 +/- 5.83 % activity as shown in Table 1)

How do the authors explain that their measured dissociation constants (Kd) are in the low nanomolar range down to even 9.03 x 10-11 (Figure 4) while the Km values are in the range of 0.54 – 1.18 g/L? Assuming a molecular weight of xanthan of very roughly 1 000 000 g/mol this would be orders of magnitude away from the Kd values. The authors should in my opinion definitely discuss the kinetic model and mechanisms in more detail. I would also suggest using the term “apparent Km” in this context.

Figure 7d: How can the “clearance” in the determination of reducing power exceed 100%?

Some experimental details (instruments used, etc.) need to be described in more detail (see my further comments).

I would strongly recommend a final language editing and English polishing.

Minors:

Introduction: I think regarding the role of the CBM it would be helpful to discuss previous experiments (truncation, engineering, etc.) that have been performed with other enzymes (glycosyl hydrolases, LPMOs, etc.).

Line 48: Grammar: change “broadly” to “broad”

Line 51: Do the authors mean “C-terminus” instead of “C terminal”?

Line 59f.: The authors write: So far, MiCBMx 59 [14] and PspCBM84 [13] were the only two reported CBMs that have been preliminarily proved to be able to identify xanthan backbone binding backbones, respectively from the endotype xanthanases PspXan9 and MiXen.” – this sentence is not clear for me. I think there is a mistake in it and I suggest rephrasing.

Line 142: I think the term “enzymolysis performance” is very unusual. Do the authors mean “catalytic efficiency”?

Line 214: replace “recombinants” by “recombinant enzymes”

Line 266: Please correct “30oC”

Line 275, 277, 286, 288: the underlined homologous sequence (as indicated by the authors) is missing

Line 309: Which Xanthan preparation was used? Manufacturer?

Line 348: The SPR equipment (type, manufacturer) and the chips used need to be specified.

Line 363: “The working temperature of all the columns was 40°C.” – Please correct grammar

Line 369: please correct grammar: “freshly cleaved mica sheets were used…”; please also specify the type/manufacturer of the mica sheets

Line 373: What do the authors mean by “landform of the sample”? I think that this term is not used correctly here.

Line 382: What do the authors mean by “distribution of xanthan digests”? Do the authors mean size distribution of resulting products?

Line 403: The term “DE value” used here is not specified in the manuscript.

Line 407: The abbreviation “DPPH” is not explained in the manuscript.

Line 413f: What equipment was used for the FTIR measurements?

Figure 463: Do the authors mean “AFM” instead of “AEM”

Author Response

Reply Letter

Title: An insight into the essential role of carbohydrate-binding modules in enzymolysis of xanthan

Authors: Xin Ni, Tong Fu, Xueyan Wang, Jingjing Zhao, Zhimin Yu, Xianzhen Li, Fan Yang

Revised manuscript: ijms-2202257

Here, we wish to take this opportunity to once again express our sincere appreciation for your constructive comments and valuable recommendations. We have carefully revised the original manuscript according to the reviewers. Also, to avoid any possible grammatical and bibliographical errors, the whole manuscript has been carefully proof-read by professional English editor James Allen, PhD, from Liwen Bianji, Edanz Group China (www.liwenbianji.cn/ac). We hope every effort from us may clear up all your confusion on this work, and satisfy you at the same time. All the answers are listed as follows:

Main concerns:
Comment 1: Structural analysis of CBMx and CBM84: The authors compare the previously published crystal structure of PspCBM84 with a I-Tasser model of MiCBMx. It would definitely be relevant to discuss the quality of the obtained model and not only the quality of the alignment. In my opinion, the predicted structure in Figure 1b indicates a rather poor quality. Furthermore, the authors state: “The secondary structure comparison showed that both CBMs contain typical 4 turns, 2 η-helixes and β-sheets (Figure 1).” This is definitely not true from the data shown. In the predicted model of MiCBMx only 6 beta sheet elements are visible to me (Figure 1b). Furthermore, the one part of the sequence of PspCBM84 that contains 3 beta sheets does not at all align with the sequence of MiCBMx. In general, I believe the authors need to revise the section on the structural properties of the CBMs. I also believe that a structural alignment of the sequences (as presented in Figure 1a) for two sequences with only 20% identity and obviously such large structural differences is generally very problematic. Additionally, the term “η-helixes” is in my opinion very unusual and I think the authors mean 310 helix by that.

Reply 1: Thanks for the critical suggestion very much. We sincerely apologize for the quality of modeling. In order to more accurately show the structure of two CBMs, we used the SWISS-MODEL (https://swissmodel.expasy.org/interactive) to predict the structure of MiCBMx. As shown in Figure 2, the predicted three-dimensional structure of MiCBMx can be aligned with that of PspCBM84 with Cα root-mean-square deviation (RMSD) of 0.073 Å over 130 out of 149 amino acids, indicating a very high structural similarity between the two CBMs. Besides, in order to better analyze the secondary structure of two CBMs, Prabi (https://npsa-prabi.ibcp.fr/cgi-bin/npsa_automat.pl?page=npsa_sopma.html) were applied to predict and compare the secondary structure of PspCBM84 and MiCBMx. Both two CBMs contain a large amount of random coil structure in addition to the common alpha helix, beta turn and extended strand structures, but no η-helix or 310 helix, and their sequence similarity is 40.82%.

All the supporting information has been rationally added in “Results and Discussion” section (page 3, lines 82-102).

Figure 2. Structure properties of MiCBMx and PspCBM84. (a) The structure of PspXan9 (6fhj), in which the blue part is the structure of PspCBM84. (b) The prediction three dimensional structure of MiCBMx. (c) Three-dimensional structure comparison of the PspCBM84 (blue) and MiCBMx (green).

Comment 2: Biochemical characterization - Figure 3: The authors used 4 different buffer systems to measure their pH-dependent activity and stability. They should indicate this in the figure and the figure caption. Not only pH value but also buffer composition could influence activity as well as stability.

Reply 2: Thanks for the reasonable advice very much. As suggest, the buffer systems used are shown in the figure and the figure caption, as follows:

Figure 4. Biochemical characteristics of MiXen-CD(■), MiXen-CBM84(●) and MiXen-CBMx(▲) [14]. (a) The optimun reaction temperatures of the endotype xanthanases with xanthan as the substrate. (b) The thermostability of the endotype xanthanases. (c) The optimum reaction pHs of the endotype xanthanases with xanthan as the substrate. (d)The pH stability of three endotype xanthanases. Buffers: HOAc-NaOAc buffer (pH 4.0 to 6.0), NaH2PO4-Na2HPO4 buffer (pH 6.0 to 8.0), Tris-HCl buffer (pH 8.0-9.0), and NaHCO3-NaOH buffer (pH 9.5 to 10.0).

All the supporting information has been rationally added in Figure 4 (page 5, lines 148-155).

Comment 3: Line 129f: The authors write “As shown in Table 1, except for Ni2+, Cu2+ and NH4+, which were inhibitors for all the three recombinants, the other metal ions activated or repressed the enzyme activities in completely different ways.” How do the authors come to this conclusion and how do they differentiate between “inhibitors” and metal ions that “repressed” enzyme activity? This is not conclusive to me. Furthermore, as NH4+ (described as inhibitor by the authors) shows an activating or at least neutral effect on MiXen-CD (103.4 ± 5.83 % activity as shown in Table 1)

Reply 3: We sincerely apologize for this mistake. Firstly, this is really our negligence, NH4+ did not repress all the enzymes. Secondly, we also apologize for our improper use of the word “inhibitors”,in deed, we want to express that “except for Ni2+ and Cu2+, which repressed the activity of all the enzymes, the effects of the other ions on the activities of the enzymes varied between the enzymes.

All the supporting information has been rationally added in “Results and Discussion” section (page 5, lines 157-159).

Comment 4: How do the authors explain that their measured dissociation constants (KD) are in the low nanomolar range down to even 9.03 x 10-11 (Figure 4) while the Km values are in the range of 0.54 – 1.18 g/L? Assuming a molecular weight of xanthan of very roughly 1 000 000 g/mol this would be orders of magnitude away from the KD values. The authors should in my opinion definitely discuss the kinetic model and mechanisms in more detail. I would also suggest using the term “apparent Km” in this context.

Reply 4: We sincerely apologize for the unclear statement. In deed, the modules that work in the two modes are different, and the calculation methods of KD and KM are also different. KD is the dissociation constant, reflecting the affinity of the compound to the target. The calculation formula is KD=Kd / Ka, Kd: dissociation rate constant, Ka: association rate constant. When using SPR to characterize the dynamic characteristics of binding, xanthan mainly combines with CBM in the ligand fixed on the chip. Then, under the action of buffer, the analyte spontaneously separated from the ligand, and the equilibrium dissociation constant KD was calculated by software. Km: Michaelis-Menten constant, is the concentration of substrate when the enzymatic reaction reaches half of the maximum reaction rate. When using the kinetic model to characterize the characteristics of the enzyme, it is mainly based on the activity values measured by the reaction of the enzyme with different concentrations of substrate at different times. The apparent Km and Vmax values are further calculated using the Michael-Menton equation. This process is mainly dominated by the cutting rate of xanthan by the catalytic domain in the enzyme. Therefore, there may be magnitude differences between the two methods, the same results has also appeared in the experiment of Yang et al [14]. In addition, due to the similarity of the main chain structure of xanthan and cellulose, the Km value of some endocellulase can be used to evaluate the endotype xanthanases, such as the cellulases from Aspergillus niger [36], Thermomyces duponti [37] and Bacillus sp. respectively [38], which Km are about 0.23 g/L-0.8 g/L, which further proves the feasibility of our kinetic model.

All the supporting information has been rationally added in “Results and Discussion” section (page 6, lines 177-181).

Supplementary references

14.Yang, F.; Li, H.; Sun, J.; Guo, X.; Zhang, X. Characterization of a novel endo-type xanthanase MiXen from xanthan-degrading Microbacterium sp. XT11. Appl. Environ. Microb, 2018,85, 1-47.

36.Sulyman, A. O.; Igunnu, A.; Malomo, S. O. Isolation, purification and characterization of cellulase produced by Aspergillus niger cultured on Arachis hypogaea shells. Heliyon, 2020, 6(12): e05668.

37.Nisar, K.; Abdullah, R.; Kaleem, A.; Lqtedar, M.; Aftab, M.; Saleem, F. Purification, characterization and thermodynamic analysis of cellulases produced from Thermomyces dupontii and its industrial applications. Saudi J Biol Sci, 2022, 29(12): 103483.

38.Sriariyanun, M.; Tantayotai, P.; Yasurin, P.; Pornwongthong, P.; Cheenkachorn, K. Production, purification and characterization of an ionic liquid tolerant cellulase from Bacillus sp. isolated from rice paddy field soil. Electron J Biotechn, 2016, 19: 23-28.

Comment 5: Figure 7d: How can the “clearance” in the determination of reducing power exceed 100%?

Reply 5: We sincerely apologized for the unclear statement. The determination method of hydroxyl radical scavenging activity is based on Fenton reaction [52], and its calculation formula is: scavenging rate (%)=(Ai-A0)/(Aj-A0)×100%, Ai is the experimental group, Aj is the control group, and A0 is the blank group. Because the calculated result is a ratio, it exceeds 100%, the same results have been obtained in the experiment of Xiong et al [9].

9.Xiong, X.; Li, M.; Xie, j.; Jin, Q.; Xue, B.; Sun, T. Antioxidant activity of xanthan oligosaccharides prepared by different degradation methods. Carbohydr. Polym, 2013, 92, 1166-1171.

Comment 6: I would strongly recommend a final language editing and English polishing.

Reply 6: Thanks for the constructive suggestion very much. To avoid any possible grammatical and bibliographical errors, the whole manuscript has been carefully proof-read by professional English editor James Allen, PhD, from Liwen Bianji, Edanz Group China (www.liwenbianji.cn/ac).

Minor Concerns:

Comment 1: Introduction: I think regarding the role of the CBM it would be helpful to discuss previous experiments (truncation, engineering, etc.) that have been performed with other enzymes (glycosyl hydrolases, LPMOs, etc.).

Reply 1: Thanks for the reasonable advice very much. As suggest, we have added the description of some experiments using CBM to modify glycoside hydrolase, as follows:

For example, by fusing with the CBM from Pseudomonas fluorescens, the catalytic efficiency of the cellulase EGE has been increased by 2-6 times [27], besides, after fusing with the CBM4-2, the recombinant enzyme AXE1dC-CBM4-2 has twice catalytic efficiency as AXE1dC [28].

All the supporting information has been rationally added in the “Introduction” (page 2, lines 57-60).

Comment 2: Line 48: Grammar: change “broadly” to “broad”

Reply 2: We sincerely apologize for this mistake. We have changed “broadly” to “broad” (page 2, line 52).

Comment 3: Line 51: Do the authors mean “C-terminus” instead of “C terminal”?

Reply 3: We sincerely apologize for this mistake. We have changed “C terminal” to “C-terminus” (page 2, line 55).

Comment 4: Line 59f.: The authors write: So far, MiCBMx [14] and PspCBM84 [13] were the only two reported CBMs that have been preliminarily proved to be able to identify xanthan backbone binding backbones, respectively from the endotype xanthanases PspXan9 and MiXen.” – this sentence is not clear for me. I think there is a mistake in it and I suggest rephrasing.

Reply 4: We sincerely apologized for the unclear statement. As suggest, we have reshaped this part as follows:

So far, Microbacterium sp. XT11 (Mi) CBMx [14] and Paenibacillus sp. 62047 (Psp) CBM84 [13], from the endotype xanthanases MiXen and PspXan9 respectively, are the only CBMs that have been shown to be able to identify xanthan backbones.

All the supporting information has been rationally added in the “Introduction” (page 2, lines 65-68).

Comment 5: Line 142: I think the term “enzymolysis performance” is very unusual. Do the authors mean “catalytic efficiency”?

Reply 5: Thank you very much for your valuable suggestions. We have changed “enzymolysis performance” to “catalytic efficiency” (page 6, line 174).

Comment 6: Line 214: replace “recombinants” by “recombinant enzymes”

Reply 6: Thank you very much for your valuable suggestions. As suggest, We have changed “recombinants” to “recombinant enzymes” (page 9, line 259).

Comment 7: Line 266: Please correct “30oC”

Reply 7: We sincerely apologize for this mistake. We have corrected this mistakes (page 12, line 327).

Comment 8: Line 275, 277, 286, 288: the underlined homologous sequence (as indicated by the authors) is missing

Reply 8: We sincerely apologize for this mistake. We have added underline to the homologous sequence.

All the supporting information has been rationally added in the “Material and Method” (page 12, lines 336, 338, 347, 349).

Comment 9: Line 309: Which Xanthan preparation was used? Manufacturer?

Reply 9: Thank you very much for your valuable suggestions. Xanthan used in this experiment was purchased from Sigma.

All the supporting information has been rationally added in the “Material and Method” (page 13, lines 370).

Comment 10: Line 348: The SPR equipment (type, manufacturer) and the chips used need to be specified.

Reply 10: Thanks for the constructive suggestion very much. We have added information about SPR equipment (type, manufacturer) and chips used in the “Materials and Methods”, as follows:

Surface plasmon resonance (SPR) measurement was performed on a BIAcore T200 instrument with a sensor chip with dextran matrix and pre-coated streptavidin (GE Healthcare).

All the supporting information has been rationally added in the “Materials and Methods” (page 13, lines 405-407).

Comment 11: Line 363: “The working temperature of all the columns was 40°C.” – Please correct grammar

Reply 11: Thank you very much for your valuable suggestions. We have corrected the grammar here “The column temperature was set to 40°C” (page 14, line 424).

Comment 12: Line 369: please correct grammar: “freshly cleaved mica sheets were used…”; please also specify the type/manufacturer of the mica sheets

Reply 12: Thank you very much for your valuable suggestions. We rearranged the language here and specify the type/manufacturer of the mica sheets as follows:

To allow imaging of individual well separated xanthan molecules, 5 μL aliquots of undigested or digested xanthan samples (10 mg/L) were pipetted onto freshly cleaved mica sheets (Park Systems, Korea) and dried under a gentle flow of dry nitrogen gas.

All the supporting information has been rationally added in the “Materials and Methods” (page 14, lines 429-431).

Comment 13: Line 373: What do the authors mean by “landform of the sample”? I think that this term is not used correctly here.

Reply 13: Thank you very much for your valuable suggestions. We rearranged the language here, as follows:

Topographical and error signal mode imaging of the samples was then conducted using a commercial AFM (XE-Bio, Park Systems, Korea) in non-contact mode with NCHR monolithic silicon cantilevers (Park Systems, Korea).

All the supporting information has been rationally added in the “Materials and Methods” (page 14, lines 432-434).

Comment 14: Line 382: What do the authors mean by “distribution of xanthan digests”? Do the authors mean size distribution of resulting products?

Reply 14: We sincerely apologized for the unclear statement. We have changed “The distribution of xanthan digests” to “The size distribution of the products of digestion of xanthan” (page 14, line 444).

Comment 15: Line 403: The term “DE value” used here is not specified in the manuscript.

Reply 15: We sincerely apologized for the unclear statement. We have added the definition of DE value in the “Materials and Methods”, as follows:

A 10-mL reaction was carried out by the standard enzyme reaction process. The yield was calculated as the DE value-the DE value represents glucose equivalents, which means that the percentage of reducing sugar in dry matter was calculated by taking all the reducing sugar in the reaction system as glucose [50].

All the supporting information has been rationally added in the “Materials and Methods” (page 14, lines 460-463).

Comment 16: Line 407: The abbreviation “DPPH” is not explained in the manuscript.

Reply 16: We sincerely apologized for the unclear statement. We have added the full name of DPPH (1,1-diphenyl-2-picrylhydrazyl) in the “Results and Discussion” and the “Materials and Methods” (page 11, line 280, page 15, line 467).

Comment 17: Line 413f: What equipment was used for the FTIR measurements?

Reply 17: We sincerely apologized for the unclear statement. We have added the model and manufacturer of the fourier-transform infrared spectrum used here, as follows:

Fourier-transform infrared spectroscopy (Nicolet iS5 FTIR ThermoFisher) was used to analyze digested xanthan.

All the supporting information has been rationally added in the “Materials and Methods” (page 15, lines 474-475).

Comment 18: Figure 463: Do the authors mean “AFM” instead of “AEM”

Reply 18: We sincerely apologized for this mistake. We have changed “AEM” to “AFM” (page 8, line 221).

Best Regards!

Fan Yang

On behalf of Xin Ni, Tong Fu, Xueyan Wang, Jingjing Zhao, Zhimin Yu, Xianzhen Li

Reviewer 2 Report

Dear Authors,

I have provided my comments in the attached file. The detailed study has been performed by authors , so there are no major revisions required. But , I have small concerns which I have mentioned in my comments.

Author Response

Reply Letter

Title: An insight into the essential role of carbohydrate-binding modules in enzymolysis of xanthan

Authors: Xin Ni, Tong Fu, Xueyan Wang, Jingjing Zhao, Zhimin Yu, Xianzhen Li, Fan Yang

Revised manuscript: ijms-2202257

Here, we wish to take this opportunity to once again express our sincere appreciation for your constructive comments and valuable recommendations. We have carefully revised the original manuscript according to the reviewers. Also, to avoid any possible grammatical and bibliographical errors, the whole manuscript has been carefully proof-read by professional English editor James Allen, PhD, from Liwen Bianji, Edanz Group China (www.liwenbianji.cn/ac). We hope every effort from us may clear up all your confusion on this work, and satisfy you at the same time. All the answers are listed as follows:

Comment 1: In this articles, the essential role of two carbohydrate binding modules MiCBMx and PspCBM84, respectively derived from Microbacterium sp. XT11 and Paenibacillus sp. 62047, against xanthan by endotype xanthanase MiXen were investigated. The studies performed is scientifically sound and informative.

Reply 1: Many thanks for your positive comments .It is my great honours receiving your recommendation.

Comment 2: The article is well written considering explanation of the subject, however there are some flaws which needs to be improved.

Reply 2: Thanks for the constructive suggestion very much. We have carefully revised the original manuscript according to the reviewers, also to avoid any possible grammatical and bibliographical errors, the whole manuscript has been carefully proof-read by professional English editor James Allen, PhD, from Liwen Bianji, Edanz Group China (www.liwenbianji.cn/ac).

Comment 3: Tilldate several studies have been published on related topic. What is the significance of your studies.

Reply 3: We sincerely answer this question. In this paper, by fusing two CBMs (PspCBM84 and MiCBMx) with endotype xanthase (MiXen-CD), respectively, we proved for the first time that the feasibility of CBMs in the rational design of endotype xanthanase. The presence of both two CBMs significantly increased the activity of endotype xanthanase and the relative content of oligoxanthan in the hydrolysate. Notably, the activity of endotype xanthanase was increased by 16 times after being fused with PspCBM84. In addition, our results also confirmed that the higher the catalytic efficiency of endotype xanthanase, the higher the content of pyruvate group in the produced xanthan digest, and the better its antioxidant activity. The results of this work lay a foundation for the rational design of endotype xanthanase and the industrial production of oligoxanthan in the future.

All the supporting information has been rationally added in the “Results and discussion” (page 10, lines 263-270, page 11-12, lines 304-309).

Comment 4: Some figures need to be improved in terms of resolution e.g. Figure A1, Fig 6.

Reply 4: Thanks for the constructive suggestion very much. As suggest, we have adjusted the pictures of the full text, rearranged some pictures and enhanced their readability.

Best Regards!

Fan Yang

On behalf of Xin Ni, Tong Fu, Xueyan Wang, Jingjing Zhao, Zhimin Yu, Xianzhen Li

Reviewer 3 Report

The article submitted for expertise is a study to improve the enzymatic affinity against xanthan, in the catalytic properties of the xanthanase MiXen endotype. The study focuses on the role of the two carbohydrate-binding modules MiCBMx and PspCBM84, respectively derived from Microbacterium sp. XT11 and Paenibacillus sp. 62047. The accessibility of enzymes to xanthan substrates is deficient, which therefore hinders the industrial production of functional oligoxanthan. The characterization study shows that MiCBMx generates higher catalytic efficiency. The presence of the two CBMs allows the xanthanase endotype to produce more oligoxanthan, and especially the xanthan digestions prepared by MiXen-CBM84 lead to better antioxidant activity due to the higher content of active oligosaccharides.

The study is complete in its methodology and the results obtained. However, the discussions of each result should be more developed and not limited to a few lines. It is important that the authors show the interest of their study in relation to work in the field. Need to clearly highlight their contributions.

The references must be improved because very few recent references apart from self-citations; need to update references.

All appendixes must be in the text. This therefore requires a new formatting.

I suggest the following corrections and some questions;

- line 28; indicate names in complete nomenclature of carbohydrates at least in this sentence. GlcUAp, no in Figure ? "D" must be written in font size smaller than the body of the text; if text in 12, write "D" in 10.

- line 29; "Figure 2a" ? You must number the figures so that the first quoted is "Figure 1".

- In "Figure2a"; it is imperative to change the representation at the level of the link inter-sugar bonds. It is necessary to adopt a representation that does not lead to any possible confusion. In classic representation, a line break indicates the presence of a "C" so with the representation indicated (Figure 2a) we "read" sugar-CH2-O-CH2-sugar! Same in Figure 6.

- line 41-43; add Table or explain more.

- line 80, 86; numbering here Figure2.

- line 99; Figure A1 ??

- line106-107; why not refine the study of the impact of temperature to the optimum and proceed to temperature effects from degree to degree? Same with pH (line 116-121).

- Figure 3; lack of readability, enlarge. Same in Figure 5

- line 128; "The effects of 10 common different metal ions ..."; explain why these choices.

- line 133; "via" replace by "via"

- line 138; "The final concentrations of the ions were 1 mM, respectively"; explain.

- Figure 4; put the title of the figure after the figure. Same with Figure 5, Figure 6, Figure 7.

- Table 2; "“-” represent the data was not displayed". That's to say; not carried out.

- line 170; "Figure A2" ?

- line 191-198; Is this really new?

- line 200; "Table A1" ?

- line 212; "Figure A3" ?

Necessary to review the numbering of the figures and to harmonize their numbering nomenclature. It is also important to respect the rules of both nomenclature and representation in terms of carbohydrates.

Author Response

Reply Letter

Title: An insight into the essential role of carbohydrate-binding modules in enzymolysis of xanthan

Authors: Xin Ni, Tong Fu, Xueyan Wang, Jingjing Zhao, Zhimin Yu, Xianzhen Li, Fan Yang

Revised manuscript: ijms-2202257

Here, we wish to take this opportunity to once again express our sincere appreciation for your constructive comments and valuable recommendations. We have carefully revised the original manuscript according to the reviewers. Also, to avoid any possible grammatical and bibliographical errors, the whole manuscript has been carefully proof-read by professional English editor James Allen, PhD, from Liwen Bianji, Edanz Group China (www.liwenbianji.cn/ac). We hope every effort from us may clear up all your confusion on this work, and satisfy you at the same time. All the answers are listed as follows:

Comment 1: The study is complete in its methodology and the results obtained. However, the discussions of each result should be more developed and not limited to a few lines. It is important that the authors show the interest of their study in relation to work in the field. Need to clearly highlight their contributions.

Reply 1: Thanks for the constructive suggestion very much. As suggest, we discussed some of the results in more detail (page 3, lines 81-96, page 4-5, lines 131-147, page 5-6, lines 157-168, page 6, lines 177-181, page 7, lines 214-219, page 10, lines 263-270, page 11-12, lines 304-309). And we also highlighted the contribution of this work. In this paper, by fusing two CBMs (PspCBM84 and MiCBMx) with endotype xanthase (MiXen-CD), respectively, we proved for the first time that the feasibility of CBMs in the rational design of endotype xanthanase. The presence of both two CBMs significantly increased the activity of endotype xanthanase and the relative content of oligoxanthan in the hydrolysate. Notably, the activity of endotype xanthanase was increased by 16 times after being fused with PspCBM84. In addition, our results also confirmed that the higher the catalytic efficiency of endotype xanthanase, the higher the content of pyruvate group in the produced xanthan digest, and the better its antioxidant activity. The results of this work lay a foundation for the rational design of endotype xanthanase and the industrial production of oligoxanthan in the future.

All the supporting information has been rationally added in the “Results and discussion” (page 10, lines 263-270, page 11-12, lines 304-309).

Comment 2: The references must be improved because very few recent references apart from self-citations; need to update references.

Reply 2: We appreciate the significant comment from the reviewer. As suggest, we have updated the references and added more recent research, and the research in the past five years has reached 50% in the references.

All the supporting information has been rationally added in the “References” (page 16-17, lines 510, 512, 519, 541, 543, 545, 548, 551, 553, 556, 559, 562, 565, 568, 579, 582, 584, 586, 588).

Comment 3: All appendixes must be in the text. This therefore requires a new formatting.

Reply 3: Thanks for the constructive suggestion very much. We have shown all the contents in the appendix in the text and redesigned some pictures, including:Figure 1, Table 1, Figure 2, Figure 3, Figure 4, Figure 6, Table 4, Figure 8.

All the supporting information has been rationally added in Figure 1 (page 2, lines 48-49), Table 1 (page 3, line 97), Figure 2 (page 3, lines 98-102), Figure 3 (page 4, lines 112-119), Figure 4 (page 5,lines 148-155), Figure 6 (page 8, lines 220-224), Table 4 (page 9, lines 244-245), Figure 8 (page 10, lines 271-275).

Comment 4: line 28; indicate names in complete nomenclature of carbohydrates at least in this sentence. GlcUAp, no in Figure ? "D" must be written in font size smaller than the body of the text; if text in 12, write "D" in 10.

Reply 4: We sincerely apologized for the unclear statement. The GlcA shown in the Figure 1 and Figure 8 is the abbreviation of GlcUAp. We have added a description in the text (page 1, line 28). Besides, as suggest, we have adjusted the font size of "D". The text uses 10 font, and "D" uses 8 font (page 1, line 28).

Comment 5: line 29; "Figure 2a" ? You must number the figures so that the first quoted is "Figure 1".

Reply 5: Thanks for the constructive suggestion very much. We have rearranged and numbered the Figures in the full text, and the Figure cited here is Figure 1 (page 1, line 29).

Comment 6: In "Figure2a"; it is imperative to change the representation at the level of the link inter-sugar bonds. It is necessary to adopt a representation that does not lead to any possible confusion. In classic representation, a line break indicates the presence of a "C" so with the representation indicated (Figure 2a) we "read" sugar-CH2-O-CH2-sugar! Same in Figure 6.

Reply 6: Thanks for the constructive suggestion very much. As suggest, we have changed the representation at the level of the link inter-sugar bonds in Figure 1 and Figure 8.

All the supporting information has been rationally added in the Figure 1, and Figure 8, (page 2, line 48, page 10, line 271).

Comment 7: line 41-43; add Table or explain more.

Reply 7: Thank you very much for your valuable suggestions. We have explained more here as follows:

At present, only two endotype xanthanases have been reported, including PspXan9, a family 9 glycoside hydrolase (GH9) from Paenibacillus sp. 62047 [13], and MiXen, also a GH9 from Microbacterium sp. XT11 [14]. The molecular weights of the two enzymes are very similar, and both two enzymes could randomly cleave glycosidic bonds within xanthan substrates, but their efficiency of cutting xanthan is significantly different [13,14]. Although PspXan9 can degrade xanthan more effectively, the highly ordered structure leads to the poor enzymatic accessibility of xanthan, which subsequently reduces the hydrolysis efficiency of endotype xanthanses [15].

All the supporting information has been rationally added in the “Introduction”, (page 1, lines 40-45).

Comment 8: line 80, 86; numbering here Figure2.

Reply 8: Thank you very much for your valuable suggestions. We have rearranged and numbered the Figures in the full text, and the Figure cited here is Figure 2 (page 3, line 82, 87, 95).

Comment 9: line 99; Figure A1 ??

Reply 9: We sincerely apologized for the unclear statement. As suggest, We have shown all the contents in the appendix in the text and redesigned some pictures, including: Figure 1, Table 1, Figure 2, Figure 3, Figure 4, Figure 6, Table 4, Figure 8, and the Figure cited here is Figure 3 (page 4, line 111).

Comment 10: line106-107; why not refine the study of the impact of temperature to the optimum and proceed to temperature effects from degree to degree? Same with pH (line 116-121).

Reply 10: We sincerely apologized for the unclear statement. As suggest, we discussed the effect of temperature and pH on enzyme in more depth, as follows:

The optimum reaction conditions and the stabilities of the three recombinant endotype xanthanases were measured when the substrate was xanthan. As shown in Figure 4a, the optimal temperature for the activity of MiXen-CBM84 was 55oC, which was significantly lower than for MiXen-CD (65oC) but higher than for MiXen-CBMx (40oC) [14]. As shown in Figure 4b, MiXen-CBM84 retained >60% of its maximun activity after incubation for 1 h at 40 to 60oC; MiXen-CBMx retained >60% of its maximun activity after incubation at 20 to 45oC, and MiXen-CD retained >60% of its activity after incubation at 40 to 50oC [14]. The high thermostability of MiXen-CBM84 is probably due to the thermotolerance of the PspCBM84 domain, which is derived from the thermophilic bacterium Paenibacillus sp. 62047 [13]. Similar investigation has reported previously that the thermal stability of a pullulanase was significantly improved by fusion with a CBM from thermophilic strain Anoxybacillus sp. LM18-11 [32]. In addition, MiXen-CBMx shows stability at a low temperature. However, when the temperature is higher than 45oC, its stability will decline rapidly, which proves that MiCBMx may not work under high temperature [14]. The optimal pHs for reaction of MiXen-CBM84 (7.5) and MiXen-CBMx (7.0) [14] were lower than that of MiXen-CD (8.0), and the optimum pHs of all the three enzymes are shown in NaH2PO4-Na2HPO4 buffer (Figure 4c). MiXen-CBM84 retained >60% of its activity after incubation for 1 h at pH 6-9, MiXen-CBMx retained >60% of its activity at pH 5-9, and MiXen-CD retained >60% of its activity after incubation at pH 7-8.5 (Figure 4d). Notably, MiXen-CBMx retained the most enzymatic activity when placed in acidic (pH 4-6.5) environments, and MiXen-CBM84 retained the most enzymatic activity when placed in alkaline (pH 8-9.5) environments, this results may be caused by the different tolerance of the two CBMs to pH, which is also consistent with Zeng et al [32]. Besides, when the pH of the buffer is higher than 9, the stability of all the enzymes will decline rapidly. Thus, both MiXen-CBM84 and MiXen-CBMx were stable at a wider pH range than MiXen-CD. These results confirmed that fusion with either CBM improved the pH stability of the endotype xanthanase.

All the supporting information has been rationally added in the “Results and discussion” (pages 5-6, lines 120-147).

Comment 11: Figure 3; lack of readability, enlarge. Same in Figure 5

Reply 11: Thank you very much for your valuable suggestions. We have enlarged Figures 3 and Figure 5 to enhance their readability (pages 5, line 148, page 9, line 238).

Comment 12: line 128; "The effects of 10 common different metal ions ..."; explain why these choices.

Reply 12: We sincerely apologized for the unclear statement. In our previous literature and method studies, many experiments to determine the effect of metal ions on enzyme activity used metal ions, such as K+, Mg2+, Mn2+, NH4+, Zn2+, Ni2+, Cu2+, Fe2+, etc. In addition, it is reported that the addition of metal ions may lead to a denser structure of xanthan to some extent, which is not conducive to the enzymatic hydrolysis of xanthanthe Therefore, we selected some metal ions that have been studied in the degradation process of xanthan. The above 10 are selected after comprehensive consideration.

Supplementary references

15.Kool, M. M.; Schols, H. A.; Delahaije, R. J.; Sworn, G.; Wierenga, P. A.; Gruppen, H. The influence of the primary and secondary xanthan structure on the enzymatic hydrolysis of the xanthan backbone. Carbohydr. Polym, 2013, 97, 368-375.

16.Gu, J.; Wang, D.; Wang, Q.; Liu, W.; Chen, X.; Li, X.; Yang, F. Novel β-Glucosidase Mibgl3 from Microbacterium sp. XT11 with Oligoxanthan-Hydrolyzing Activity. J. Agr. Food. Chem, 2022, 70, 8713-8724.

48.Guan, F.; Han, Y.; Yan, K.; Zhang, Y.; Zhang, Z.; Wu, N.; Tian, J. Highly efficient production of chitooligosaccharides by enzymes mined directly from the marine metagenome. Carbohydr Polym, 2020, 234, 115909.

Comment 13: line 133; "via" replace by "via"

Reply 13: We sincerely apologized for the mistake. We have rewritten this part, as follows:

This investigation suggests that xanthanases with different CBM domains might recognize and bind different metal ions which disrupt or stabilize the conformation of the enzyme and lead to changes in the catalytic activity [33] (page 6, lines 161-163).

Comment 14: line 138; "The final concentrations of the ions were 1 mM, respectively"; explain.

Reply 14: We sincerely answer your question. In our previous literature and method research, a lot of experiments to determine the effect of metal ions on enzyme activity use metal ions with a concentration of 1 mM. Besides, it has been reported that the addition of higher concentrations of metal ions will, to some extent, lead to a denser structure of xanthan, which is not conducive to the enzymatic hydrolysis of xanthan. Therefore, we chose to control the concentration of metal ions at a relatively low concentration of 1 mM.

Supplementary references

14.Yang, F.; Li, H.; Sun, J.; Guo, X.; Zhang, X. Characterization of a novel endo-type xanthanase MiXen from xanthan-degrading Microbacterium sp. XT11. Appl. Environ. Microb, 2018,85, 1-47.

15.Kool, M. M.; Schols, H. A.; Delahaije, R. J.; Sworn, G.; Wierenga, P. A.; Gruppen, H. The influence of the primary and secondary xanthan structure on the enzymatic hydrolysis of the xanthan backbone. Carbohydr. Polym, 2013, 97, 368-375.

16.Gu, J.; Wang, D.; Wang, Q.; Liu, W.; Chen, X.; Li, X.; Yang, F. Novel β-Glucosidase Mibgl3 from Microbacterium sp. XT11 with Oligoxanthan-Hydrolyzing Activity. J. Agr. Food. Chem, 2022, 70, 8713-8724.

47.Zhang, Z.; Tang, L.; Bao, M.; Liu, Z.; Yu, W.; Han, F. Functional characterization of carbohydrate-binding modules in a new alginate lyase, TsAly7B, from Thalassomonas sp. LD5. Marine drugs, 2019, 18, 25.

48.Guan, F.; Han, Y.; Yan, K.; Zhang, Y.; Zhang, Z.; Wu, N.; Tian, J. Highly efficient production of chitooligosaccharides by enzymes mined directly from the marine metagenome. Carbohydr Polym, 2020, 234, 115909.

Comment 15: Figure 4; put the title of the figure after the figure. Same with Figure 5, Figure 6, Figure 7.

Reply 15: We sincerely apologized for the mistake. As suggest, we have put the title of the figure after the figure (Figure 4, Figure 5, Figure 6, Figure 7) (page 5, line 148, page 7, line 196, page 8, line 220, page 9,line 238).

Comment 16: Table 2; "“-” represent the data was not displayed". That's to say; not carried out.

Reply 16: We sincerely apologized for the mistake. We have removed this part to avoid confusion.

Comment 17: line 170; "Figure A2" ?

Reply 17: We sincerely apologized for the unclear statement. As suggest, We have shown all the contents in the appendix in the text and redesigned some pictures, including: Figure 1, Table 1, Figure 2, Figure 3, Figure 4, Figure 6, Table 4, Figure 8, and the Figure cited here is Figure 6.

Comment 18: line 191-198; Is this really new?

Reply 18: We sincerely apologized for the unclear statement. To avoid any doubt, we have added the molecular mass corresponding to each peak in this section to better distinguish them, as follows:

As shown in Figure 7a, the original xanthan had a high molecular weight (3 to 4×106 Da) and a retention time (Rt) of 18-21 min. After incubation with the respective endotype xanthanases, in addition to residual xanthan (Rt =18-21 min) that was not degraded, some products with intermediate molecular weight (6 to 9×105 Da) (Rt =21-24 min) appeared in the samples.

All the supporting information has been rationally added in the “Results and discussion” (page 8, lines 229-234).

Comment 19: line 200; "Table A1" ?

Reply 19: We sincerely apologized for the unclear statement. As suggest, We have shown all the contents in the appendix in the text and redesigned some pictures, including: Figure 1, Table 1, Figure 2, Figure 3, Figure 4, Figure 6, Table 4, Figure 8, and the Table cited here is Table 4.

Comment 20: line 212; "Figure A3" ?

Reply 20: We sincerely apologized for the unclear statement. As suggest, We have shown all the contents in the appendix in the text and redesigned some pictures, including: Figure 1, Table 1, Figure 2, Figure 3, Figure 4, Figure 6, Table 4, Figure 8, and the Figure cited here is Figure 8.

Best Regards!

Fan Yang

On behalf of Xin Ni, Tong Fu, Xueyan Wang, Jingjing Zhao, Zhimin Yu, Xianzhen Li

Round 2

Reviewer 1 Report

The authors have improved the manuscript and answered all the points I raised, made corrections or added information where appropriate. I only would like to propose some minor changes:

Line 84: “Among all the models given, MiCBMx and PspCBM84 show the highest similarity and their sequence similarity is also 40.82%” Do the authors mean that they received the model with the best score in Swiss model using PspCBM84 (6fhj) as template? Please clarify and rephrase.

Line 99: What are the purple, green and cyan spheres showing? Are they representing metal ions? If yes, are the metal ions artifacts of the crystallization procedure or structural features? Please add some description in the Figure caption.

Author Response

Reply Letter

Title: An insight into the essential role of carbohydrate-binding modules in enzymolysis of xanthan

Authors: Xin Ni, Tong Fu, Xueyan Wang, Jingjing Zhao, Zhimin Yu, Xianzhen Li, Fan Yang

Revised manuscript: ijms-2202257

Here, we wish to take this opportunity to once again express our sincere appreciation for your constructive comments and valuable recommendations. We have carefully revised the original manuscript according to the reviewers. We hope every effort from us may clear up all your confusion on this work, and satisfy you at the same time. All the answers are listed as follows:

Comment 1: Line 84: “Among all the models given, MiCBMx and PspCBM84 show the highest similarity and their sequence similarity is also 40.82%” Do the authors mean that they received the model with the best score in Swiss model using PspCBM84 (6fhj) as template? Please clarify and rephrase.

Reply 1: Thanks for the critical suggestion very much. We sincerely apologize for the unclear statement. When SWISS MODEL was used to predict the structure of MiCBMx, the system only gives two prediction results. The first one was the structure predicted using the crystal structure of PspCBM84 (6fjh.1.A) from Paenibacillus sp. 62047 as the template, and the sequence similarity between MiCBMx and PspCBM84 can reach 40.82%, while the other one used the crystal structure of Lprg (3mh8.2.A) from Mycobacterium Tuberosis as the template, but the sequence similarity between MiCBMx and this structure was only 10.81%, therefore, the MiCBMx structure predicted with PspCBM84 as the template was selected for subsequent analysis.

All the supporting information has been rationally added in “Results and Discussion” (page 3, lines 84-89).

Comment 2: Line 99: What are the purple, green and cyan spheres showing? Are they representing metal ions? If yes, are the metal ions artifacts of the crystallization procedure or structural features? Please add some description in the Figure caption.

Reply 2: Thanks for the reasonable advice very much. To avoid any confusion, we redraw Figure 2, and as suggest, the description of the purple and cyan spheres showing in Figure 2 has added in the Figure caption, as follows:

Figure 2. Structure properties of MiCBMx and PspCBM84. (a) The structure of PspXan9 (6fhj), in which the blue part is the structure of PspCBM84. (b) The prediction three dimensional structure of MiCBMx. (c) Three-dimensional structure comparison of the PspCBM84 (blue) and MiCBMx (green). The calcium ions are shown as cyan spheres, sodium as purple [13].

All the supporting information has been rationally added in “Results and Discussion” (page 3-4, lines 100-105).

Best Regards!

Fan Yang

On behalf of Xin Ni, Tong Fu, Xueyan Wang, Jingjing Zhao, Zhimin Yu, Xianzhen Li

Reviewer 3 Report

The authors having answered all my questions and my suggestions; I therefore propose the publication of this article.

Author Response

Many thanks for your positive comments .It is my great honours receiving your recommendation.